# Hardware and Software Implementation of the Embedded Controlling System for the TuMag Camera

**Manuel Rodríguez Valido** [1,*], **David Orozco Suárez** [2], **David Hernández Exposito** [3], **Eduardo Magdaleno Castelló** [1] **and Basilio Ruiz Cobo** [3]

1   Department of Industrial Engineering, University of La Laguna, 38203 San Cristóbal de La Laguna, Spain; emagcas@ull.edu.es
2   Institute of Astrofísica de Andalucía, Consejo Superior de Investigaciones Científicas, 18080 Granada, Spain; orozco@iaa.es
3   Institute of Astrophysics of Canary Islands, 38205 San Cristóbal de La Laguna, Spain; davidhdeze@gmail.com (D.H.E.); brc@iac.es (B.R.C.)
*   Correspondence: mrvalido@ull.edu.es

**Abstract:** The main objective of this paper is to present the design and implementation of an embedded controlling system for the Tunable Magnetograph (TuMag) instrument camera sensor and test-bench software application for camera laboratory characterization. The TuMag camera is based on a new scientific sensor GPIXEL SENSE400 device family equipped with an FPGA. The camera sensor has an active area of 2048 × 2048 pixels and reaches a frame rate of up to 48 frames per second. The embedded controlling system was implemented on an Artix 7 FPGA, which oversees controlling the image sensor through a configuration interface. It controls the read-out of the sensor data and the communication of commands with the host device. The camera has a standard CoaXPress communication interface of up to 3125 Gbps. The FPGA embedded control system allows bit, word, and serial channel calibration and the modification of several parameters, such as the region of interest (RoI) size, exposition time, hardware or software trigger, single or continuous acquisition modes, sensor gain, and adjustment of the black level offset and exposition time. In addition, the firmware has a temperature controller, voltage level monitoring, and an enable–disable power sensor. In addition, the test-bench software application is a library developed in Python 3.7. It is a wrapper for the standard GeniCan commercial frame grabber interpreter. This wrapper permits the improvement of the interaction and interface between the user and the hardware in the camera calibration. The test-bench software application permits the reduction of costs and the transport risk of the TuMag camera between different laboratories. All TuMag embedded control systems and test-bench software application functionalities were successfully tested according to scientific requirements.

**Keywords:** FPGA; embedded system; camera sensor; spectropolarimeter

## 1. Introduction

The Sunrise III balloon-borne solar mission [1], led by the Max Planck Institute for Solar System Research (MPS; Germany) and with partners from the USA, Japan, Spain, and Germany, is the third solar observatory dedicated to the investigation of key processes governing the physics of magnetic fields and convective plasma flows in the lower solar atmosphere. These processes are crucial for our understanding of the magnetic activity of the Sun and the outwards transport of energy to heat its outer atmosphere and to fuel eruptions and coronal mass ejections, phenomena that affect the Earth's environment. Designed for operation in the stratosphere (at heights of approximately 37 km) to avoid image degradation due to turbulence in the lower terrestrial atmosphere, Sunrise III carries three scientific instruments: the UV Spectropolarimeter and Imager (SUSI) instrument, a grating-based spectropolarimeter that will explore the rich near-UV range between 300 nm and 410 nm [2]; the Chromospheric Infrared spectroPolarimeter (SCIP), which allows the

combined measurement of photospheric and chromospheric layers using spectral lines formed in the infrared [3]; and the Tunable Magnetograph (TuMag), a visible imaging spectropolarimeter able to observe the solar photosphere and the lower chromosphere [4], which was fully developed and built by the Spanish Space Solar Physics Consortium (S$^3$PC) (S$^3$PC partners are IAA-CSIC, IAC (Instituto de Astrofísica de Canarias), IDR/UPM (Instituto de Microgravedad "Ignacio Da Riva", Universidad Politécnica de Madrid), INTA (Instituto Nacional de Técnica Aeroespacial), and UV (Universidad de Valencia)) led by the Instituto de Astrofísica de Andalucía (IAA-CSIC, Spain).

The design and development of a novel instrumentation for stratospheric flights is a technological challenge due to the near-vacuum conditions prevailing at those heights. In the case of SCIP and TuMag instruments, scientific cameras are one of the most critical subsystems. Indeed, none of the available cameras in the market met the scientific requirements of the instruments in terms of photon budget, speed, and flight compatibility. Hence, the S$^3$PC designed and developed new cameras from scratch [5], except for the sensor, which was purchased on the market. In particular, the TuMag camera is based on the scientific sensor GPIXEL SENSE400-BSI [6–8], a rolling shutter, backside illuminated sensor with an active area of 2048 × 2048 pixels that reaches a frame rate of up to 48 frames per second in standard mode (STD).

The main objective of this paper is to present the design and implementation of an Embedded Controlling System for the TuMag camera sensor (TuMag-ECS) and a Test Bench Software Library, TBSL, to manage thermal characterization in the laboratory. The main task of the TuMag-ECS is to manage the 3125 Mbps CoaXPress Communication interface between the sensor and the host as well as to control any other IP or custom module to run and configure the camera and monitor the housekeeping of the device. The TBSL, on the other hand, is a useful tool for optical and thermal characterization tests of the instrument under vacuum conditions. That is, before the flight, rigorous and demanding processes will be carried out to validate the performance of the TuMag camera in near-vacuum conditions. Consequently, the TuMag instrument, including the camera and the host (the full optical and electronics units), must be transported to each center to mount different experimental setups. The transport of this equipment involves risks to its integrity and requires human and economic project resources to execute the necessary logistics. For these reasons, we have developed the TSBL, which is based on a commercial CoaXPress frame grabber, to be able to test the TuMag camera in different laboratories of the consortium of the research group without the need to carry the whole instrument.

The design presented in this work represents a challenge and an advancement in terms of scientific and technical knowledge. The TuMag-ECS design allows us to investigate the behavior of different technologies and methods, such as the Gpixel CMOS detector, the Artix 7 FPGA, and the CoaXPress interface, in a harsh environment.

The rest of the paper is organized as follows: Section 2, Materials and Methods, describes the general FPGA block architecture of the TuMag camera. This section details the TuMag-ECS architecture and its main hardware blocks and functionalities, as well as the embedded software that controls the communications between the camera–host and the test bench software application to set up laboratory experiments. Finally, Section 3 presents the results and conclusions.

## 2. Materials and Methods

The architecture of the TuMag camera FPGA blocks diagram is shown in Figure 1. It consists of two main blocks: the *sensor driver* module and the *TuMag embedded control system* module. The *sensor driver* module is responsible for direct communication with the CMOS detector. It defines an *SPI* interface to access the internal registers of the sensor, a finite state machine circuit that generates 19 timing control signals to read the sensor data synchronously, and a circuit to read and convert 8 serial LVDS 12-bit data channels to 12 parallel words. In addition, the *sensor driver* has implemented a hardware circuit to calibrate the sensor, channel order, readout electronic control, etc. The *TuMag Embedded*

*Control System* (TuMag-ECS) module is a core-based MicroBlaze soft processor that manages the 3125 Gbps CoaXPress communication interface [9] between the host (or frame grabber) and the sensor. It also serves as a communication interface between the heater and the temperature controller sensor, monitors the supply 24 V voltage of the TuMag camera, and switches the sensor on/off. Additionally, the TuMag-ECS, through the AXI-Lite and AXI4 video stream buses, sets and manages the communications of the *sensor driver* to configure the sensor and/or read the data. The TuMag camera firmware was implemented in an Artix 7 FPGA using VHDL and Verilog hardware description language. The Artix 7 FPGA device provides the highest performance-per-watt fabric, transceiver line rates, and DSP processing in a cost-optimized FPGA [10].

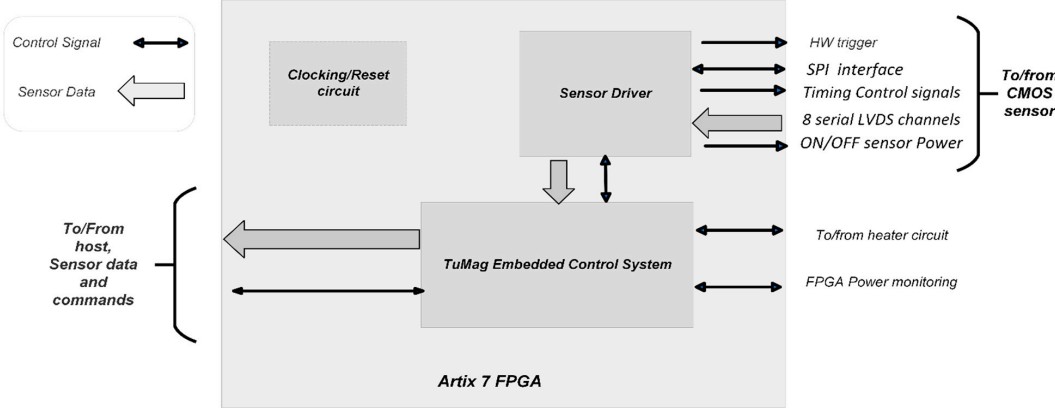

**Figure 1.** TuMag camera FPGA blocks diagram.

In addition, TuMag-ECS is in charge of bit, word, and serial channel calibration and of the modification of several parameters, such as the region of interest (ROI) size, exposure time, hardware or software trigger, single or continuous acquisition modes, sensor gain, and adjustment of the black level offset.

The sensor GSENSE400BSI is a scientific [6–8], back-illuminated CMOS image sensor with a wide dynamic range of pixels produced by Gpixel INC. The sensor has the advantages of high sensitivity, low noise, and high integration, including a 12-bit ADC, a temperature sensor, a PLL, and an SPI controller on-chip. The ADC is used to convert analog signals into 12-bit digital signals, and the PLL is used for on-chip clock management. The SPI controller is used for on-chip register configuration, which can change the on-chip PGA gain, data readout direction, PLL crossover multiplier, and operating mode. In addition, the CMOS sensor has two modes of operation: standard (STD) mode and high dynamic range (HDR) mode. Table 1 lists the key specifications of GSENSE400BSI.

**Table 1.** List of the key specifications of GSENSE400BSI [11].

| Parameter | Value |
|---|---|
| Optical format | 2.0 inch |
| Active image size | 22.528 × 22.528 (mm) |
| Pixel size | 11 × 11 μm |
| Number of active pixels | 2048 (H) × 2048 (V) |
| Shutter types | Electronic rolling shutter |
| Frame rate | 24 fps @ HDR mode48 fps @ STD mode |
| Full well capacity | 90 ke |
| Temporal dark noise | 1.6 e |
| Dark current | $0.2\,e^-/s/pix@-50\,°C$ |
| Output format | 8 pairs of LVDS drivers |
| Package | 115 pins |
| Power Consumption | <650 mW |
| Chroma | Mono |

## 2.1. TuMag-ECS Architecture

The TuMag-ECS was implemented with the Vivado tool (Figure 2); in the Supplementary Material, we added a PDF file with the block diagram of all TuMag-ECS circuits. The main task of this system is to control both the GSENSE400BSI *sensor driver* (right side) and the CoaXPress communication interface with the host or frame grabber (left side). In addition, it also controls/monitors other modules (IP) necessary for the instrument, such as the *heater* module (temperature controller), the *SPI* module to read the internal sensor registers, the *train* module to put the sensor in training mode, the *AXI_to Camera register* necessary to configure the sensor, the *XADC* module to monitor internal FPGA parameters, and, finally, the *Easii_cxp_dev* controller of the CoaXPress communication interface.

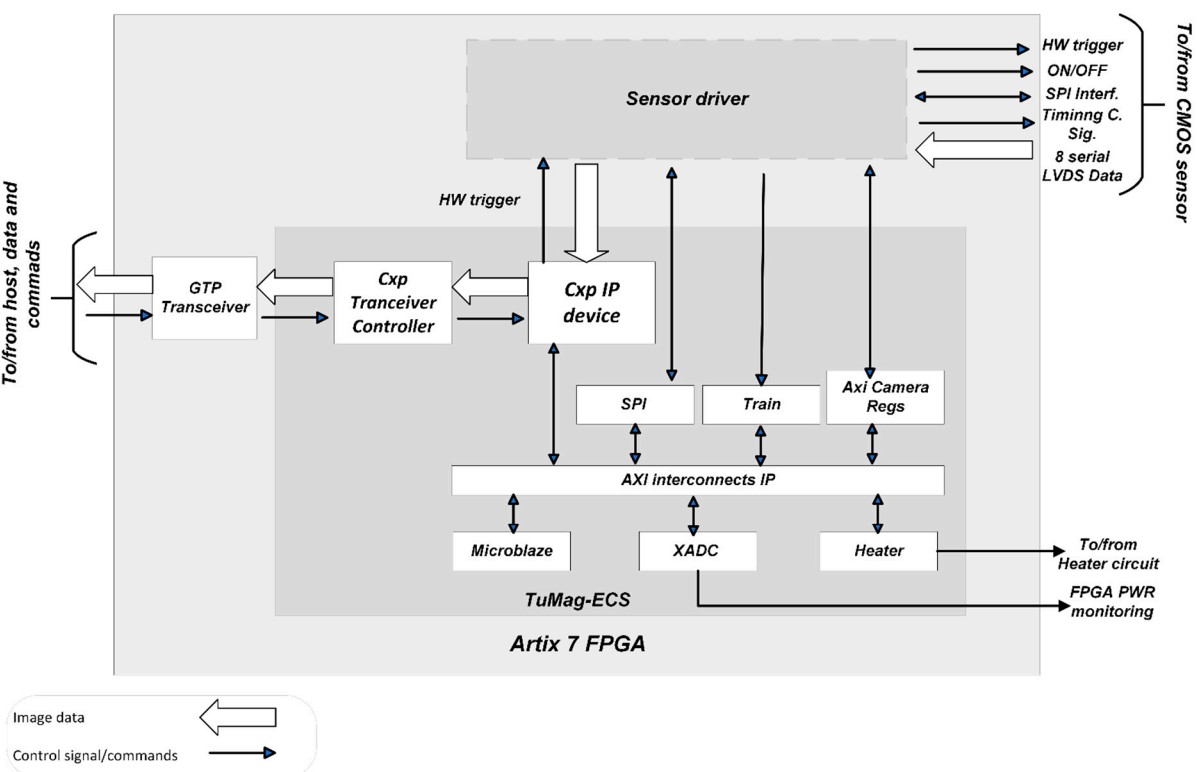

**Figure 2.** TuMag-ECS blocks diagram architecture. The AXI Interconnect manages the AXI transactions between the AXI Master (MicroBlaze) and the AXI Slaves modules.

This embedded system is based on an architecture defined by an AXI4-Lite bus, which, by means of the software drivers of each module, allows carrying out the necessary reading and writing operations for the following tasks: monitoring the heater status, monitoring the supply voltage of the system, configuring the operating modes of the sensor, and managing all other functionalities defined in the firmware of TuMag ECS. These drivers are implemented through simple software functions executed on a MicroBlaze soft-processor IP core (2017.1) provided by Xilinx. In this processor, the software control application controls/decodes the communications between the sensor and the host and vice versa. Additionally, it manages all IPs belonging to the architecture of the embedded control system. These IP cores are the *heater* control, *spi*, *AXI_to_camerareg*, sensor *train*, *PWR*, *Cxp device IP*, and *Cxp Transceiver controller*. The latter is connected to the FPGA *GTP transceiver* interface.

### 2.1.1. CoaXPress Communication Interface

The data transfer from the CMOS sensor to the instrument host is performed via a CoaXPress interface. This interface connects devices (typically cameras) to a host (frame grabber) and allows, over the same cable (single coaxial cable), data transfers plus com-

mands and power. The CoaXPress standard is a fast, point-to-point asymmetric speed (20.833 Mbit/s to uplink and up to 6 Gbit/s to downlink) 8B/10B encoded packet serial communication [12,13]. On each connection, the link protocol defines a set of logical channels, which are used to send trigger signals, general-purpose inputs and outputs, control data, and high-speed streaming video transmission. Each device connected to the interface is controlled through the control channel by the host.

Figure 3 shows the CoaXPress interface modules (*Cxp IP device* and *CxP Transceiver Controller*). These connect the image sensor data from the CMOS sensor (Figure 2, right side) with the host or frame grabber (Figure 2, left side). Figure 3a shows the interface ports of the *Cxp device* module. It has a control AXI4-Lite interface, which is used to monitor and control all user memory space, and an AXI4 slave video stream interface dedicated to image data transfer from the TuMag *sensor driver* to the host. Table 2 shows an interface port description. The uplink and downlink data interface is connected to CoaXPress through the Artix 7 FPGA GTP transceiver and *Cxp Transceiver Controller* module, see Figure 3b. Table 3 shows interface port descriptions. The output of the GTP transceiver is connected to physical media through a Microchip chip [14].

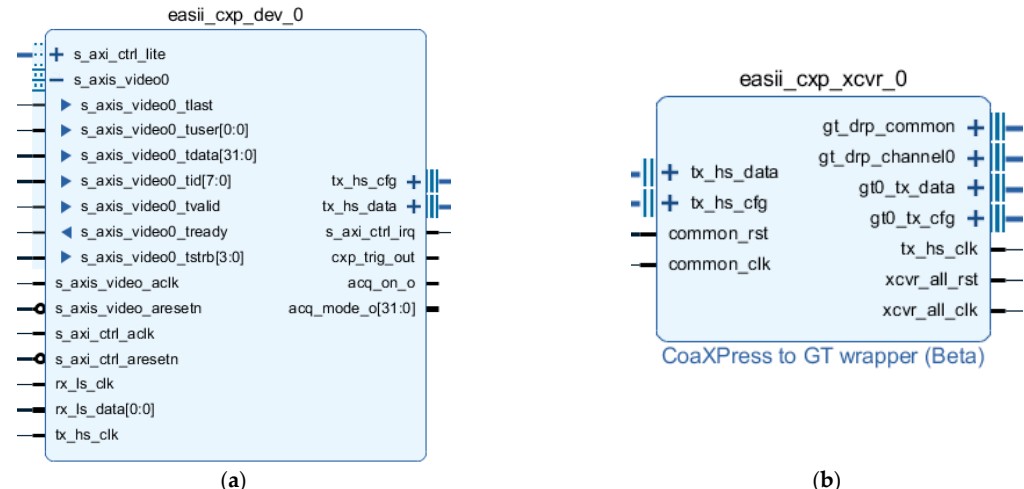

(**a**)  (**b**)

**Figure 3.** Block diagram of the embedded CxP control interface. (**a**) CoaXPress IP controller; (**b**) wrapper interface to the GTP FPGA transceiver.

**Table 2.** Port descriptions of the CoaXPress IP controller.

| Port Name | I/O Port | Description |
|---|---|---|
| s_axi_ctrl_lite | I/O Bus | AXI4-Lite interface containing write and read channels |
| s_axis_video0 | I/O Bus | AXI video data channel |
| s_axis_video0_tlast | I | Last data for current line |
| s_axis_video0_tuser | I | Start of image information |
| s_axis_video0_data | I | CXP pixel-packed data bus |
| s_axis_video0_tid | I | TID of the current stream |
| s_axis_video0_tvalid | I | Data valid information from master |
| s_axis_video0_tready | O | Slave ready to accept data |
| s_axis_video0_tstrb | I | Pixel/Metadata information |
| s_axis_video_ack | I | AXI4-Stream common clock, 125 MHz |
| s_axis_video_aresetn | I | AXI4-Stream common synchronous active-low reset |
| s_axis_ctrl_aclk | I | Clock input—no special frequency restriction |
| s_axis_ctrl_ aresetn | I | Active low reset input |
| rx_ls_clk | I | Fixed 125 MHz clock input |
| rx_ls_data | I | 20.833 Mbps data from equalizer |
| tx_hs_clk | I | SerDes clock 125 MHz |
| tx_hs_cfg | I/O Bus | SerDes configuration bus |
| tx_hs_data | I/O Bus | Uplink and Downlink port |
| s_axi_ctrl_irq | O | Synchronous rising edge interrupt |
| cxp_trig_out | O | Hw trigger from host |

**Table 3.** Port descriptions of the wrapper interface to the GTP FPGA transceiver.

| Port Name | I/O Port | Description |
|---|---|---|
| tx_hs_cfg | I/O Bus | SerDes configuration bus |
| tx_hs_data | I/O Bus | Uplink and Downlink port |
| common_ rst | I | Common Axi reset |
| common_clk | I | 50 MHz Common Axi clk |
| gt_drp_common | I/O Bus | Dynamic reconfiguration port Configuration |
| gt_drp_channel0 | I/O Bus | Dynamic reconfiguration port, channels |
| gt0_tx_data | I/O Bus | Gtp transmitter data interface |
| gt0_tx_cfg | I/O Bus | Gtp transmitter configuration interface |
| tx_hs_clk | O | SerDes clock from 125 MHz |
| xcvr_all_rst | O | Rest to GPTtransceiver |
| xcvr_all_clk | O | 100 Mhz to GPT transceiver |

Additionally, the CoaXPress interface supports the Generic Interface for Cameras or GeniCam standard [15]. It permits decoupling industrial camera interface technology such as CMOS sensor GSENSE400BSI from the user application programming interface (API). This standard support is conducted through an XML file stored in the device. This file is shared with the host at the time of connection and thus provides a description (namespace given by the standard) of the GeniCam API-compliant device (GenAPI). The boot records, stored on the *Cxp device*, provide the frame grabber with a way to access the XML file.

The TuMag-ECS CoaXPress communication interface has been tested using a commercial frame grabber installed on a PC (Euresys Coaxlink Quad frame grabber). Through it and for debugging purposes only, we have checked the operation of all the camera functionalities.

2.1.2. Heater Module

This module (Figure 4a) is necessary for the thermal control of the CMOS Sensor. Basically, a Peltier/heater device is included in the sensor PCB. The thermal control is performed by means of 4 commands managed by the host. These commands are sent from the frame grabber or host and the software of the TuMag embedded control system decodes them into local actions like setting the temperature setpoint value, enabling and disabling the heater, or reading the heater status. The *heater* control IP core configures, via SPI, the Texas Instruments DAC8311IDCKT [14] (Table 4). The SPI clock is limited to a maximum frequency of 1 MHz due to path restrictions. The *heater* core has a timer configured to periodically (every second) refresh the DAC to prevent it from losing its configuration and desired temperature and set point value.

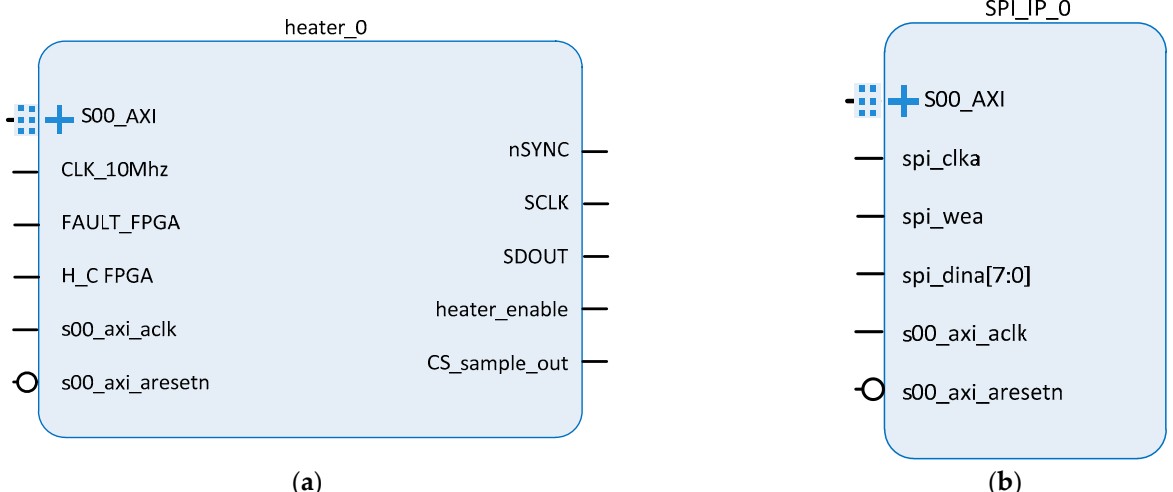

**Figure 4.** (**a**) Block diagram of the interface of the heater module. (**b**) Block diagram of the interface of the SPI module.

**Table 4.** Port descriptions of the heater module.

| Port Name | I/O Port | Description |
|-----------|----------|-------------|
| S00_AXI | I/O Bus | AXI-Lite Interface |
| CLK_10 Mhz | I | 10 MHz clock |
| FAULT_FPGA | I | Heater Status. '1' fault, '0' heater status is correct |
| H_C_FPGA | I | Heater operation. '1' warming up, '0' heater cooling |
| s00_axi_aclk | I | 50 MHz main axi clock |
| s00_axi_aresetn | I | Asynchronous low level clear |
| nSYNC | O | DAC8311, enable signal |
| SCLK | O | DAC8311 signal clock |
| SDOUT | O | Serial data signal (set point value) connect to Din of DAC8311 |
| heater_enable | O | enable/disable Heater system |
| CS_sample_out | O | Periodical Output signal for debug purpose |

### 2.1.3. SPI Module

For the GSENSE400 BSI CMOS detector, after the power-up or reset sequence, the default values for all on-chip registers are '0'. These registers should be re-programmed. The sensor driver module (Figure 1) has an integrated FSM (finite state machine) to perform read and write (re-programming) operations on the 256 bits of the internal register. Both operations are performed at a frequency of 25 MHz coming from the detector. These read-and-write sequences cannot be interrupted by each other, i.e., all 256 bits have to be read or written without interruption.

In the write operation, the FSM takes the data from the RB2 register and reprograms the internal register of the sensor. In the read operation, FSM reads the data from the sensor (organized as 32 8-bit registers) and stores it in the SPI module. The architecture of this IP is based on a dual real port BlockRam. It is wrapped with an AXI4-Lite interface on the MicroBlaze side and an 8 data bit on the *SPI* sensor side (Table 5). In this way, the memory decouples the clock domain of the sensor driver from the clock domain of the AXI4-Lite interface. Figure 4b shows the IP interface.

**Table 5.** Port descriptions of the SPI module.

| Port Name | I/O Port | Description |
|-----------|----------|-------------|
| s00_axi_aclk | I | 50 MHz main axi clock |
| s00_axi_aresetn | I | Asynchronous low level clear |
| S00_axi | I/O | AXI-Lite Interface |
| spi_din [7..0] | I | spi data input |
| spi_clk | I | 25 Mhz spi clk |
| spi_we | O | SPI write enable signal |

### 2.1.4. Train Module

The CMOS sensor has a train operation mode that permits alignment/calibration of the incoming 12-bit 8-channel serial data from the sensor. Training is run at the start of the firmware embedded software or upon request of the host. This training task is performed in the *sensor driver* block (Figure 1) and monitored by the *train module* (Table 6). When bit, word, and channels alignment is achieved, a status report is generated and sent to the instrument host. The training is repeated until all data channels are calibrated. Figure 5a depicts the *train module* interface.

**Table 6.** Port descriptions of the train module.

| Port Name | I/O Port | Description |
|---|---|---|
| s00_axi_aclk | I | 50 MHz main axi clock |
| s00_axi_aresetn | I | Asynchronous low level clear |
| S00_AXI | I/O | AXI-Lite Interface |
| train_dina [7..0] | I | spi data input |
| train_clk | Input | 25 Mhz spi clk |
| train_wea | I | Spi write enable signal |
| training_done | I | Input from training module |
| HW_TRAIN | O | Output to host hw training |

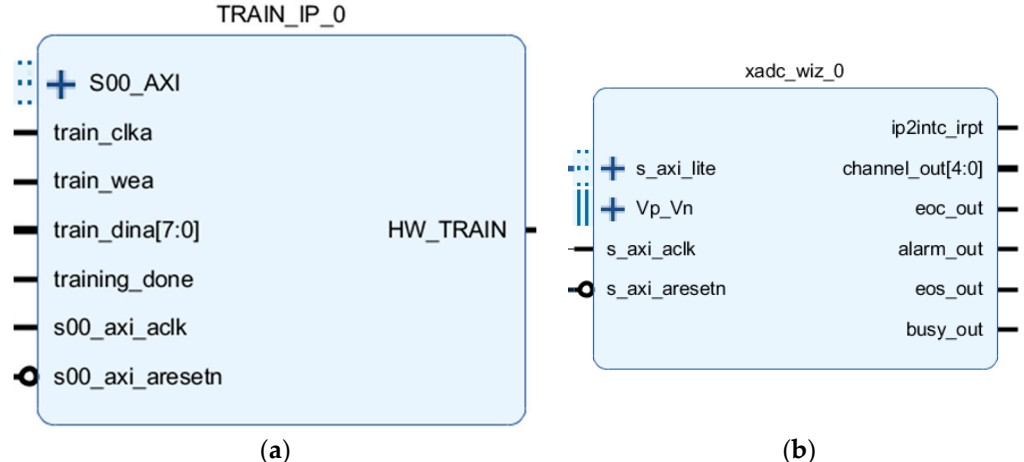

(**a**)          (**b**)

**Figure 5.** (**a**) Block diagram of the *train* module; (**b**) module to monitor 24 V circuit power.

### 2.1.5. XADC Module

This IP (inside of FPGA device) was used to monitor the internal FPGA on-chip register values such as on-chip temperature reg, on-chip VCCINT data reg, on-chip VCCAUX data reg, ADC out of VP/VN, on-chip VREFP data reg, on-chip VREFN data reg, and VBRAM data (Table 7). Figure 5b shows the IP block port interface. These measurements need to be known to be able to properly correct camera frames from dark current or gain fluctuations. Figure 5b depicts the XADC module interface.

**Table 7.** Port descriptions of the module to monitor 24 V circuit power.

| Port Name | I/O Port | Description |
|---|---|---|
| s00_axi_aclk | Input | 50 MHz main axi clock |
| s00_axi_aresetn | Input | Asynchronous low level clear |
| S00_axi | I/O | Axi4-Lite Interface |
| Vp_Vn | I | External differential input voltage |

### 2.1.6. AXI_to_Camera Register Module

The embedded software application configures and controls the operation of the CMOS through the *AXI_to_Camera register* module (Figure 6). To do this, the module has three register memory banks, RB1, RB2, and RB3, for storing the configuration parameters of the sensor driver. The RB1 register bank is used to control the data acquisition process, set the sensor in calibration, acquisition, or test mode, store the calibration words, and store the words for testing the sensor in test mode. The RB2 register bank stores the 256 bits sensor register for configuration through the FSM circuit integrated in the sensor driver module. RB3 is a register bank used by the image decoder module (in the sensor driver block) to generate the timing control signal for image acquisition. Table 8 shows the interface ports description of the camera register banks module.

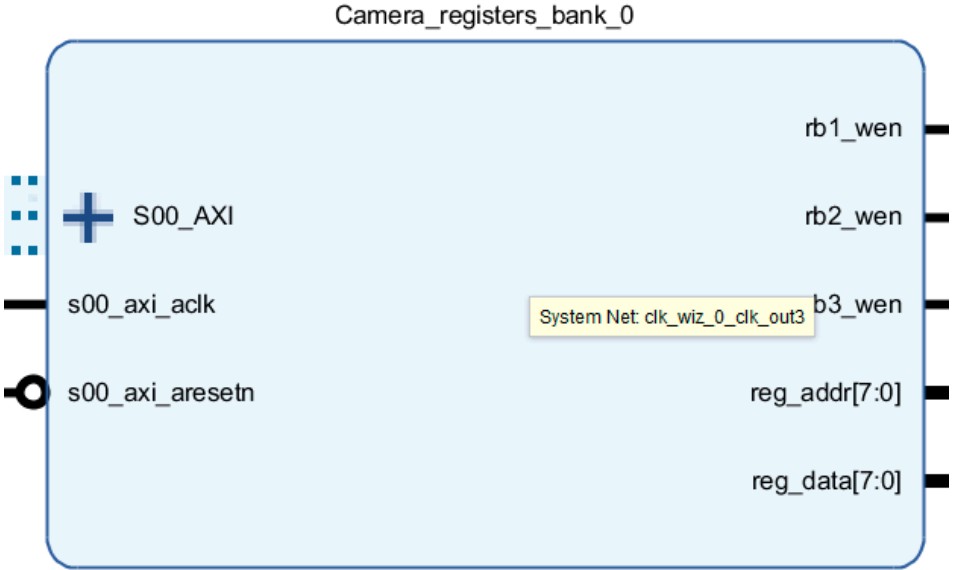

**Figure 6.** Block diagram of camera registers memory bank.

**Table 8.** Port descriptions of the camera register memory bank module.

| Port Name | I/O Port | Description |
| --- | --- | --- |
| s00_axi_aclk | Input | 50 MHz main processor axi clock |
| s00_axi_aresetn | Input | Asynchronous low level clear |
| S00_AXI | I/O | AXI-Lite Interface |
| rb1_wen | output | Signal enable/select to write data in bank 1 |
| rb2_wen | output | Signal enable/select to write data in bank 2 |
| rb3_wen | output | Signal enable/select to write data in bank 3 |
| reg_addr [7:0] | output | Bank Address to write data |
| reg_data [7:0] | output | data to be written into register banks |

### 2.2. TuMag Embedded Software Application

The TuMag camera has a set of hardware registers through which it is possible configure the acquisition process, as well as the conditions under which it is conducted. In addition, the camera has another set of registers that allow the monitoring/controlling of other system parameters, such as the FPGA input voltage or the internal sensor temperature.

The embedded application software was developed using Xilinx SDK IDE in the C programming language and under the GeniCam standard. This application, which runs continuously on the MicroBlaze soft processor, has the functionalities of initializing local FPGA registers into driver module and sensor configurations FPGA register banks (RB1, RB2, and RB3), establishing the communications system link and speed between the frame grabber (host) and the device for data acquisition (sensor), updating (to shared) the XML file (GeniCam camera standard specification) on the host, setting the interrupt map in the embedded system memory, and decoding the commands coming from the host.

Figure 7 depicts a simplified flow diagram of the main code. We can differentiate two stages: the initialization stage (Figure 7 to the left) and the CoaXPress link-command decoding stage (Figure 7 to the right).

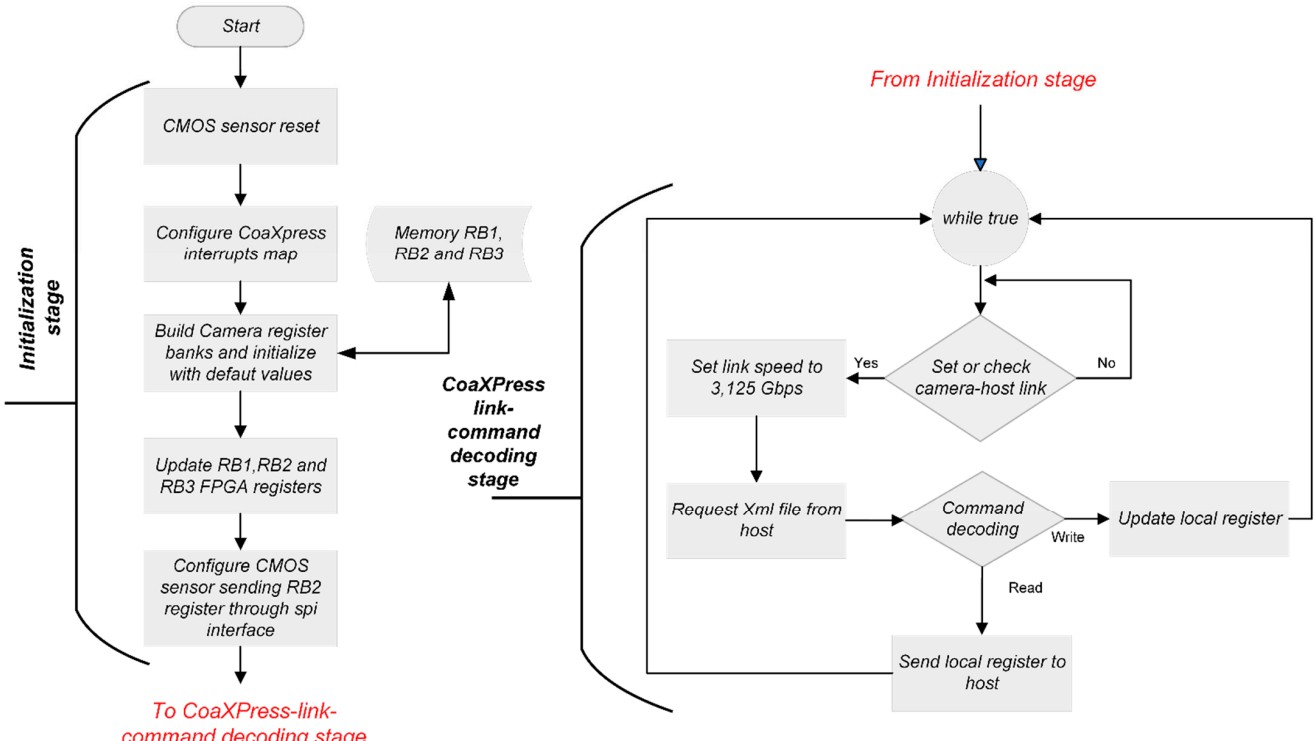

**Figure 7.** Flow diagram of embedded application software.

In the initialization stage, the application sets the CMOS sensor to the reset state, configures the CoaXPress interrupts map on MicroBlaze memory, builds and initializes the camera register banks (RB1, RB2, and RB3) on memory with default values, updates the RB1, RB2, and RB3 FPGA register into sensor driver modules, and configures the CMOS sensor with default values of the RB2 register.

In the CoaXPress link-command decoding stage, the application goes into an infinite listening loop where the device–host is linked through a discovering process, where the communication speed is set to 3125 Gbps, and where the GeniCam XML file format of the TuMag camera is exchanged with the host. This XML file, defined under the rules of the GeniCam standard, defines the configuration parameters, i.e., features of and commands of the TuMag camera, and permits the decoupling of sensor technology from the host application. The XML file, compressed in zip format, used in this design is organized in three categories: device control, image format control, and acquisition control. Within each of these categories there are different entries or registers. Each register, defined according to the GeniCam standard, has a name, an address, an access mode (Read only or Read/Write), a data type (command, integer, float, string, enumerated), and valid or allowed values. Table 9 shows some input values in the three categories.

The command decoding algorithm works like a "look at table". In the MicroBlaze memory, all GeniCam entries used are defined with their name, address, and value. The software listens for an interruption coming from the host. When this happens, four parameters are read from the CoaXPress device, the device identifier, the command (*cmd*), the address (*addr*), and the size of the message (*size*). From the value of the *cmd*, the algorithm decides whether it is a write or read operation.

**Table 9.** XML entries for some camera features.

|  | Xml Entry Name Address | Value | Access Mode | Type |
|---|---|---|---|---|
| **Device control category** |  |  |  |  |
|  | DeviceVendorName@0x2000 | EASii-IC | RO | String |
|  | DeviceFirmwareVersion@0x2090 | 1.2r16 | RO | String |
| **Image Format Control category** |  |  |  |  |
|  | WidthMax@0x3000 | 2048 | RO | integer |
|  | width@0x3024 | Default 2048 | RW | integer |
|  | PixelFormat@0x3014 | Mono16 | RW | Enumerated |
| **Acquisition Control Category** |  |  |  |  |
|  | AcquisitionStart@ 0x300C | 1 | RW | Command |
|  | AcquisitionStop@ 0x3010 | 0 | RW | Command |
|  | ExposureTime @0x3028 | Default 300 ms | RW | Float |

In the first case (writing $cmd = 0 \times 01$, hexadecimal format), the *data* are read from the CoaXPress device y with the *addr* variable, and the "look at table" is searched to update the parameter (i.e., image) with the *data* value. The write-host commands permit updating the local FPGA registers RB1, RB2, and RB3, setting the temperature value of the heater circuit, configuring the internal CMOS registers, running the sensor training process, setting the camera in acquisition (continuous, single frame or CoaXPress) or test mode, and setting the sensor power ON/OFF.

In the second case or read command, *cmd* equals $0 \times 00$, the variable address is used to read the parameter and store variable *data* to be sent to the host (i.e., *addr* = $0 \times 3028$ data < −300 ms). Just like the write operation, the host can access the FPGA parameter values.

In both cases, write and read operations, the algorithm command decoding handles the errors through an *ack* variable in the following way. By default, the value of the variable is an invalid address or FALSE. If the decoded address is valid, the *ack* variable is updated as a valid address or TRUE. At the end of the address decoding process, the *ack* variable is sent with the appropriate value.

In the camera observation mode or CoaXPress mode (24 frames per second), image acquisition is controlled directly by the host. The principle of operation of the acquisition process is as follows: the host sends the trigger command and the CoaXPress device (Figure 2) generates the *HW trigger* signal and sends it to the sensor driver.

This signal is used to start or enable an FSM that controls the sensor readout cycle. This cycle consists of three phases: resetting rows, waiting for the configured exposure time, and eventually reading out in row order [14]. Figure 8 depicts a VHDL hardware simulation of the acquisition process in the sensor driver of Figure 2, showing only 32 rows for the compressed view. After the HW trigger signal is received, the sensor driver module controls the acquisition process by generating the image request signal (*frame_req set to high* red signal in Figure 8) and generating the time signals to synchronize the data readout (blue signals in Figure 8). The *sync_x* signal sets the data reading rate on the GPIXEL sensor device and, when the *fval* signal is set to 1, controls the time period where the data are available to be transmitted to the host. After the last line, the *lval* signal is transmitted, the *grab_done* signal (red in Figure 8) is set to high, and the sensor driver control goes into an idle state, waiting for a new HW trigger or command from the host.

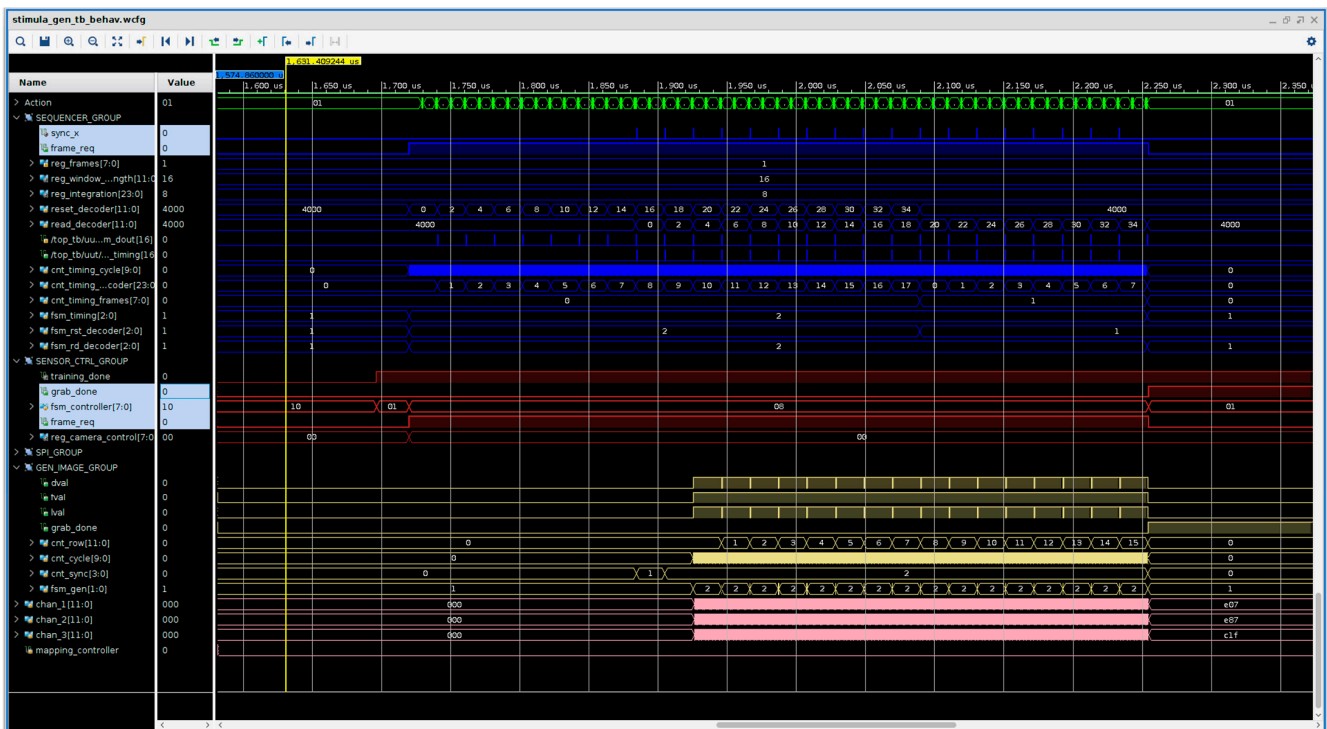

**Figure 8.** Waveform diagram of simulation frame request from the host. We only show the 32-line readout to illustrate the process of acquiring an image.

### 2.3. Test Bench Software Application

Each flight and spare camera are later subjected to the following test campaign to ensure its functionality:

1. Optical characterization under ambient conditions (∼25 °C and 1 bar).
2. Optical and electrical characterization under near-vacuum conditions and three temperatures (hot, nominal, and cold). This provides confidence that the camera behaves correctly if there is no degradation with respect to the first step. No early component failures ("infant mortality") have arisen from that first stress.
3. A bake-out for the elimination of all particles that could degrade the optics of the instrument. The camera is not yet supplied but kept at 60 °C for 72 h and 5 mbar of pressure.
4. A thermo-vacuum test (three temperatures: hot, nominal, and cold) to verify that the camera temperatures meet the thermal requirements (image sensor, FPGA, and cold finger temperatures are read by the camera itself; the temperature sensors placed on the camera cover are read by the test setup).
5. Finally, a second optical characterization gives us confidence in its behavior, before delivering the camera for assembly into the instrument.

The TuMag cameras (Figure 9) are a critical subsystem of the instrument. To avoid risks in the transport of the complete instrument between the different laboratories where the testing campaigns were carried out, we designed and implemented a test-bench software application (TBSA) to handle the TuMag camera from a local computer. This allowed us to simplify the number of devices to be transported and reduced the assembly time of experimental setups. Due to the fact that each laboratory of the research consortium has this tool at its disposal, we only moved the camera and its associated electronics between the different laboratories to be characterized.

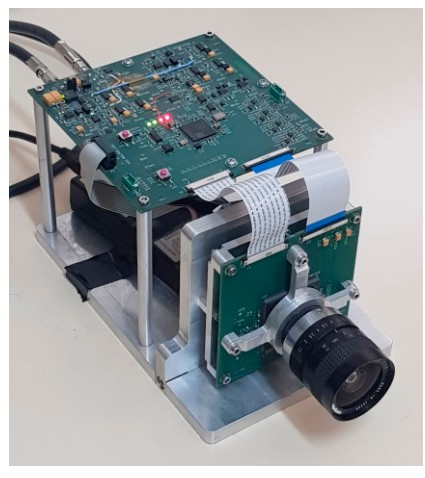

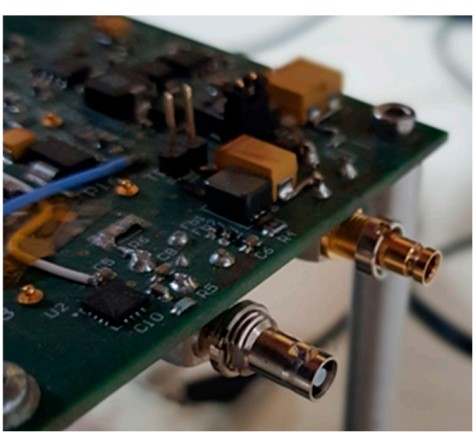

(**a**)

(**b**)

**Figure 9.** (**a**) Photograph of a TuMag camera laboratory prototype. (**b**) Photograph of the connectors of the TuMag prototype for data transmission and power input.

This TBSA, formed by a set of methods developed in Python and Java scripts, handles the Coaxlink Quad card, a trade frame grabber provided by Euresys Company installed on a personal computer. This card permits the streaming of images and configures, controls, triggers, and powers multiple high-speed cameras via CoaXPress 1.1 interface technology and GenICam programming interface (Figure 10).

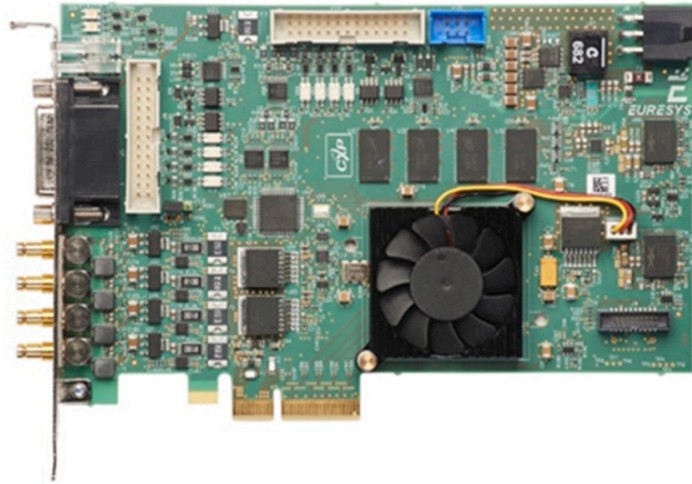

**Figure 10.** Photograph of a CoaXPress frame grabber used for the test-bench software application.

Euresys Company provides its own GenAPI script language (like JavaScript) for programming modules and scripts to handle the features of frame grabbers and cameras based on GenICam. To adapt this generic software architecture to the TuMag camera, we implemented the *tumagLib.js* and the *featuresHandler.js* scripts. The first file defines a set of functions implemented to read, write, and execute the TuMag features. The second file contains the program used to handle the instrument. Figure 11 shows a block diagram representing the hardware and software architecture of TBSA used to handle the TuMag camera.

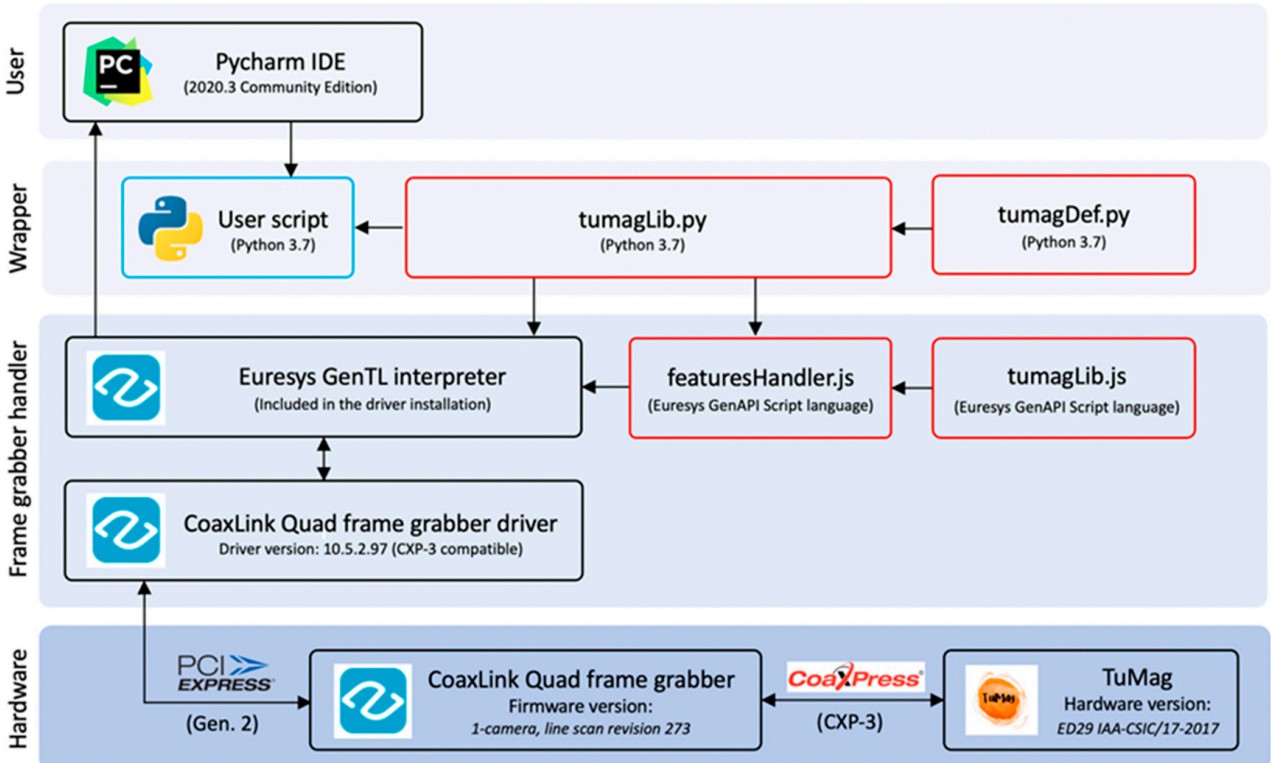

**Figure 11.** Block diagram of the hardware and software architecture, including the Python wrapper for TBSL.

When the *featuresHandler.js* is launched, the frame grabber and the TuMag camera are automatically detected first, and then the script is executed. To write, read, and execute different TuMag features, this script must be modified to add the corresponding functions defined in the tumagLib.js module. For this, the user can use any text editor. In particular, the featuresHandler.js script is executed using the GenTL Console tool, which includes a GenTL interpreter specific to the Euresys GenAPI script language.

In addition, to improve the interaction and interface between the user and the hardware, we have developed a wrapper library developed in Python. Specifically, the wrapper consists of two files: *tumagDef.py* and *tumagLib.py* (Figure 11). The first one contains a list of the function names that are defined in the *tumagLib.js* module. It also includes the definition of paths to the necessary program files and required opcodes to generate system instructions automatically. The second file contains a set of functions to access and configure each of the available TuMag features. These Python functions automatically modify the script contained in *featuresHandler.js* to add the appropriate code and directly execute the corresponding system instruction to launch the Euresys GenTL interpreter.

This Python library is formed by the get, set, and execute functions, as well as initlib. Table 10 shows the Set/Get actions of the library grouped by functionalities. We have developed a user guide (the document is available on request from the authors) where the installation of the library, CoaXPress card and drive installation, and examples of the utilization can be found.

The results are displayed in the terminal console. As an example of use, Figure 12 shows a block diagram of steps to measure Dark Image Characterization. The measurement process, defined in the Python script, is a succession of commands sent by the host to be decoded by the TuMag camera. It ends when all the gains and integration times are executed. Acquired images are sent to the host as they are captured.

**Table 10.** Main functionalities of the implemented library.

|  | Set | Get |
|---|---|---|
| Heater functions | Set temperature value of heater, and enable or disable the heater module | Get the actual status of de heater module |
| Training functions | Enables or disables training | Get the result of the training process. For each of the 8 available channels |
| Sensor configuration functions | Set region of interest (ROI), gain, mode, integration time, sampling frequency, and set if multiple or single frame capture | Get internal sensor register values, get ROI, Gain, integration time, and get sensor temperature |
| Acquisition process execution functions | Executes image acquisition, i.e., continuous acquisition (Start/Stop) and acquisition via HW trigger or via SW trigger. | Images are sent to the host/frame grabber |
| Miscellaneous functions | Enables or disables sensor power ON/OFF and camera reset | Get internal FPGA register values: temperatures and voltages, get embedded hardware and software version |

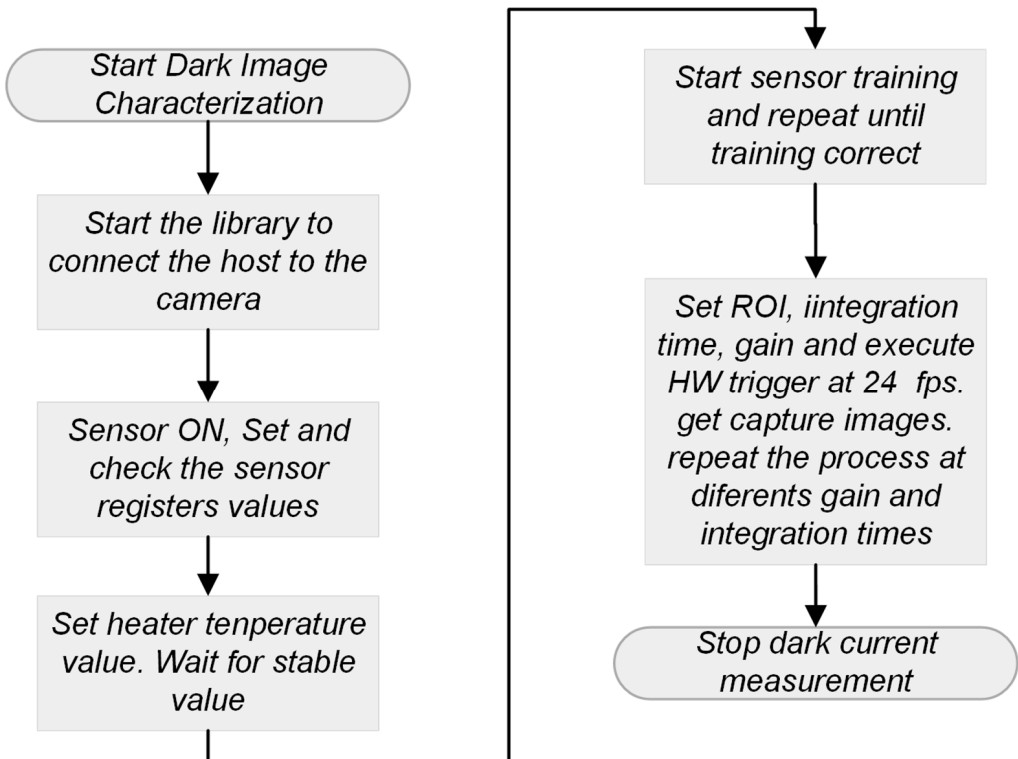

**Figure 12.** Simplified block diagram of the pipeline commands to characterize.

## 3. Results

In this paper, we have presented the implementation of a specifically tailored FPGA architecture for the TuMag camera embedded control system and test-bench software application for carrying out thermal TuMag camera tests under controlled laboratory conditions. Both developments comply with the CoaXPress and Genicam standards. The TuMag-ECS was designed using HDL languages and implemented in an Artix-7 XC7A50T-2CSG325C FPGA device. The design was completed using Vivado and SDK 2017.4 for simulation, debugging, and implementation. The design was successfully integrated into the sensor driver board FPGA and is fully operational.

The TuMag-ECS in combination with software handles communication with the host for both commands and data transfer (24 frames per second) over a CoaXPress communications interface up to 3125 Gbps. In addition, TuMag-ECS manages bit, word, and serial channel camera calibration, configuration of internal detector registers such as region of

interest (RoI) size, exposure time, hardware triggering, single or continuous acquisition modes, sensor gain, and black level offset adjustment. Also, TuMag-ECS manages the control temperature circuit of the sensor PCB, monitors FPGA parameters, and switches the sensor power ON/OFF.

The test-bench software application was implemented and tested. The TBSA tool, in addition to avoiding risks and costs in transporting the instrument, permits the reduction of the setup time of the different characterization processes on the TuMag Camera. All camera functionalities, architecture modules, and sensor configurations were successfully tested according to scientific requirements using a commercial frame grabber with a CoaXPress interface.

From the point of view of resources consumed by the Artix 7 FPGA (Table 11), most of them are below 25%, except for input/output operations which require 54%, the BUFG global clock buffers (56%), and the Mixed-Mode Clock Manager MMCM (80%). In these conditions, the employed FPGA still has enough resources to implement additional functionalities for camera frame pre-processing before the data are sent to the host.

**Table 11.** FPGA resource utilization for TuMag-ECS architecture.

| Component | Used/Available | Used (%) |
|---|---|---|
| LUT | 4847/32,600 | 15% |
| LUTRAM | 278/9600 | 3% |
| FF | 6023/65,200 | 9% |
| BRAM | 18/75 | 24% |
| IO | 81/150 | 54% |
| GT | 1/4 | 25% |
| BUFG | 18/32 | 56% |
| MMCM | 4/5 | 80% |

The camera measurements can be divided into three clearly defined stages: the conversion of photons into electrons, the transformation of electrons into voltage, and, ultimately, the digitization of this voltage. Each of these stages introduces measurement noise, the source of which must be thoroughly comprehended to effectively address or reduce its impact [5]. The characterization of the TuMag camera noise was carried out using two tests, Dark Image Characterization and Linearity and Gain Assessment. In the first one, there was a sequence of dark images (5000) at different sensor gains. And the second one, also at different sensor gains, was illuminated with LED light (530 nm). For each gain setting, a sequence of 30 different exposure times was recorded, incrementally increasing from 0.0205 milliseconds up to the point of reaching the sensor's full well capacity, which is dependent on the specific gain mode.

Table 12 shows a summary of the inferred noise parameters for different multiplicative sensor gains, where $g$ is the gain or digital numbers by electrons, $N_r$ is the readout noise due to the conversion of photons into electrons, and $\Delta/12$ is the quantization noise, the quantization interval is $\Delta = D_r/2^n$, with $D_r$ being the dynamic range (the pixel full well expressed in DN), $f_{prnu}$ is the photon response non-uniformity associated with the sensor manufacturing, $N_{dc,fpn}$ represents noises associated to dark current, that is, *shot noise* and *fixed pattern noise* resulting from pixel-to-pixel differences on manufacturing materials and alike, $L$ is the linearity or measure of the dispersion of the fits, and $S_{DC}$ is the signal due to dark current. A few dust particles were identified in the regions exhibiting anomalous low signal levels. The number of pixels covered by dust particles remained below 0.001% of the whole image for all the cameras.

**Table 12.** Noise parameters obtained from the prototype TuMag camera.

| Sensor Gain | g (DN/e$^-$) | $N_r$ (DN) | $\Delta/12$ (DN) | $f_{prnu}$ (%) | $N_{dc,\ fpn}$ (%) | L (%) | $S_{DC}$ (DN) |
|---|---|---|---|---|---|---|---|
| 1.85× | 0.0520 | 1.77 | 50,711 | 1.67 | 0.49 | 0.20 | 149.16 ± 0.68 |
| 2.49× | 0.0707 | 2.13 | 42,236 | 1.74 | 0.43 | 0.22 | 149.22 ± 0.70 |
| 3.68× | 0.1034 | 2.37 | 37,899 | 1.87 | 0.18 | 0.34 | 149.14 ± 0.82 |
| 1.29× | 0.0369 | 1.61 | 55,943 | 1.59 | 0.70 | 0.10 | 148.25 ± 0.74 |
| 3.70× | 0.1028 | 2.66 | 33,828 | 1.86 | 0.40 | 0.39 | 149.22 ± 1.10 |

Figure 13 shows the results of acquiring a 2048 × 2048 image frame with the TuMag prototype laboratory camera. The USAF resolution test chart was used as a target.

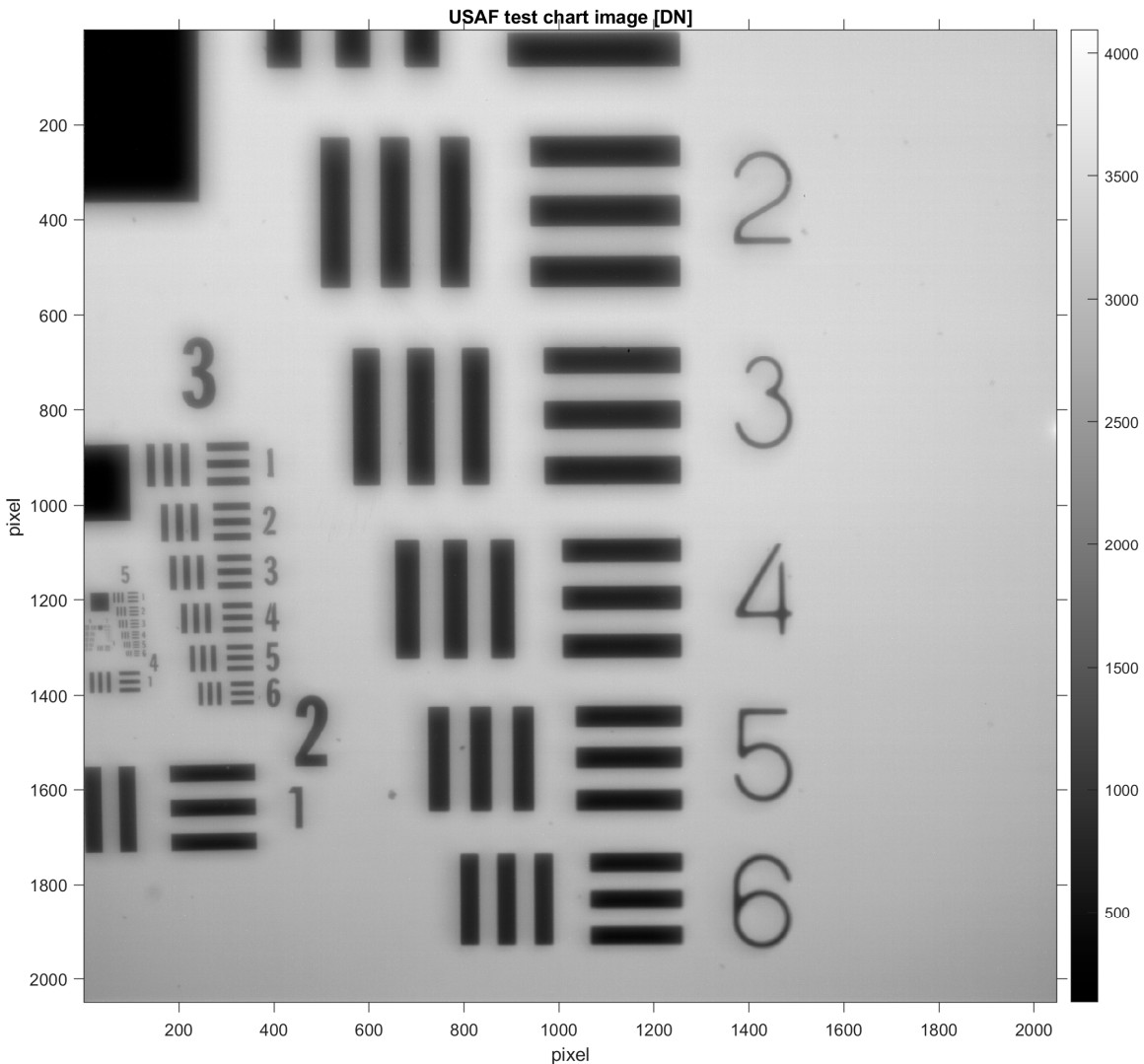

**Figure 13.** Acquired image in digital number (DN) using a USAF pattern.

Figure 14 depicts a histogram of the image signal in non-illuminated conditions, with an exposition time of 600 ms, and under vacuum and temperature-controlled conditions (25 °C). The mean value (149 DN) represents the dark current; it represents a constant offset of the image signal.

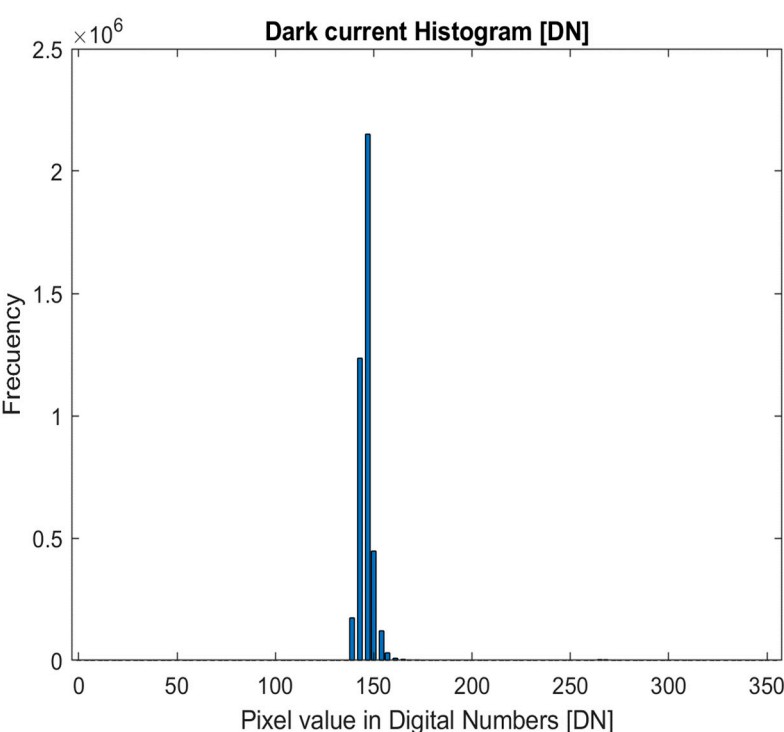

**Figure 14.** Sensor dark current histogram in digital number (DN).

## 4. Conclusions

Merely conducting a market survey is often inadequate for finding a camera that meets the required specifications. Scientific demands typically diverge significantly from commercial needs [11,16]. This has indeed been the case during the initial stages of conceptualizing our two instruments for the SUNRISE III mission.

By conducting comprehensive tests and hardware and software design, a thorough understanding of the TuMag camera's operational capabilities and performance characteristics was achieved, facilitating its effective deployment in scenarios resembling the SUNRISE III mission conditions. Also, the satisfactory results obtained in the battery of tests performed on TuMag show that the technologies used can be exploited for the design and implementation of new solar instruments. Such is the case that this embedded system and test-bench software application has also been used in the implementation of the SCIP (Sunrise Chromospheric Infrared spectroPolarimeter) instrument [5].

The GSENSE400-BSI detector is a relatively new scientific image sensor with a high dynamic range, high sensitivity, and low noise. The designs, based on this image sensor, found in the literature [17,18] are usually based on other communication interfaces such as camera links or USB. Our design used a CoaXPress communications interface with better mechanical and electrical performance (a simple coaxial cable to data and commands simplifies the connection to the host and signal-to-noise reduction). Considering that the TuMag camera must be in vacuum conditions, a serial connection, between camera and host, is more efficient than a parallel connection [19].

The attainable signal-to-noise ratio within a system subjected to illumination conditions wherein shot noise, directly proportional to the square root of the signal magnitude, predominates as the primary contributor to measurement interference is of particular significance. Consequently, the aim is to mitigate the influence of other noise sources that scale linearly with the signal magnitude. This mitigation strategy is accomplished by ensuring that the detector's filling level remains below fifty percent of its capacity.

**Supplementary Materials:** The following supporting information can be downloaded at: https://www.mdpi.com/article/10.3390/electronics12194071/s1. The block diagram of all TuMag-ECS circuits.

**Author Contributions:** Conceptualization, D.O.S., M.R.V. and D.H.E.; methodology, M.R.V., D.H.E. and E.M.C.; software, M.R.V.; validation, D.O.S., D.H.E., M.R.V. and E.M.C.; formal analysis, M.R.V., D.O.S. and D.H.E.; investigation, M.R.V., D.O.S. and E.M.C.; writing—original draft preparation, M.R.V. and D.O.S.; writing—review and editing, M.R.V. and D.O.S.; supervision, D.O.S., M.R.V. and B.R.C. All authors have read and agreed to the published version of the manuscript.

**Funding:** This research was funded by the RTI2018-096886-B-C53 Project of the Ministry of Science.

**Institutional Review Board Statement:** Not applicable.

**Informed Consent Statement:** Not applicable.

**Data Availability Statement:** Not applicable.

**Conflicts of Interest:** The authors declare no conflict of interest.

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
