# Peer review of "Hardware and Software Implementation of the Embedded Controlling System for the TuMag Camera"

_electronics, doi:10.3390/electronics12194071_

Round 1

Reviewer 1 Report

The paper brings the interesting design of the tunable magnetograph camera interface system for space operation. The implementation target is the FPGA  device Artix-7. As it is illustrated, the interface integrates the camera image interface utilizing 8-lane high-speed serial links. The camera unit is controlled by the SPI interface. The designed interface system simplifies interfacing to the camera for the host system. It uses a wide-band serial link for data transfer and control interface. To minimize the design effort the control over the designed interface has been passed to MicroBlaze embedded microcontroller/microprocessor. 

The first problem encountered in the paper is the bus system organization. Figure 2 shows the block diagram that suggests that the AXI bus links in parallel to all modules. The AXI bus system utilizes pear to pear connection model so the figure requires modification as well the description. There should be marked master and slave parts of the AXI interface and respective AXI bridges and arbitration modules The tables that gather a detailed list of signals in the AXI interface are not necessary. It is enough to state the precise type of AXI  protocol and slave/master interface side. As it was stated the AXI-Lite is utilized that implements single data packet transfers. Similarly AXI. When some signals are not connected they should not be mentioned at all (not used). 

There are several illustrations of developed modules (IP cores). From the point of view of the paper, it is not necessary as the detailed block diagram is better for understanding the authors' ideas and design partitioning. I strongly recommend improving the block diagrams as not all shown IP modules are placed on block diagrams. 

The other problem encountered in the description is a multiplicity of clock signals. After paper analysis, it can be observed that there is a need of delivering a 150MHz clock signal that can be the source and the reference for other signals. From this signal can be easily derived the 25MHz clock assuring a synchronous relationship between clock signals. The synchronous approach to the designed system would reduce the clock domain crossing problems as well as minimize the problem of data flow at the boundaries of clock domains. It is recommended to analyze the documents explaining the use of the clock to enable signal to reduce operating frequency. 

The implementation summary is usually given in the number of particular resources used instead of its percentage of the whole circuit. Next, it can be referred to as the total number of resources in the selected chip. I strongly recommend giving detailed resource usage as a table that illustrates the number of LUTs further divided into usage details like logic (LUT), distributed RAM, SRL, number of flip-flops and BlockRAM units (possibly the DSP48 if used with utilized configuration multiplier, accumulator adder). The separate resource usage information should be given to the MicroBlaze microcontroller as it is not user designed part.

The attention requires the implementation of a portable test bench. Analysing the paper it is difficult to imagine what the portability means for the authors. What ideas have been used to deliver the easy portable camera interface enabling further satellite system development without passing the delicate camera?

The command channel described here uses an interesting XML-based idea. The terminology used here as an XML file does not reflect the usage way of the channel. The command conversation utilizes the XM:L formatting (like HTML for web browser). It is a good idea to show part of such command conversation developed for the needs of the system. Some explanation is required about command parsing and error handling in the command stream.

The programmatic part is an interesting connection of multiple pieces of hardware and software. It is difficult to understand JavaScript and Python usage. The questionable is switching between different languages in a way based on script generation and later its execution. It would be profitable for the design to write the Python library that directly interacts with the frame grabber driver.

Summarizing the paper brings interesting design implementation of satellite camera interface along with ideas of using a portable testbench for fast development avoiding transfer of delicate camera unit. The paper requires several improvements of the presented material to emphasize interesting details and authors' ideas while removing unnecessary redundancy in the IP cores presentation. 

Please find additional remarks enclosed in the manuscript.

The paper requires improvement in the presentation. The enclosed manuscript contains marked sentences requiring the author's attention. There can be found several mistakes in naming e.g.: GTP is mistyped as "gpt".

Other remarks are placed on the enclosed manuscript.

Author Response

Dear reviewer,

Thank you for your effort and accurate comments. We have written in red text the answer to your comments.

Reviewer 2 Report

The paper presents an embedded system on FPGA to control a Tunable Magnetograph (TuMag) camera. The proposed system is composed of an FPGA-based embedded controller designed with HDL and a testbench software designed in Python to handle the frame grabber. I think the work needs a very deep revision, since in the present form it is not clear how the system works and how it improves the state-of-the-art. I suggest the following revisions:

1) The system is shown in Section 2 with the HDL modules presented as black boxes with the input/output pins. I think the authors should also present how the HDL modules are interconnected and present some HDL simulations to make clear how the system works.

2) In Section 2.3 the authors discuss how the test bench software application helps to avoid moving the TuMag optical and electronic units between different laboratories. I do not understand how this is achieved. This part must be discussed in more details.

3) The section on the experimental results must be much improved with data from the experiments carried out using the proposed system.

4) The authors have placed the experimental results and the conclusion in a single section (section 3). I think the conclusions must be presented in a separate section.

5) The authors should discuss how the proposed system improves the state-of-the-art by presenting the system performance and comparing it with similar systems from literature.

6) Some errors and typos must be corrected, such as: line 110 “.” is missing; in table 1 the value of dark current is missing; line 160 the character “?” must not be present; line 173 Figure 2b is probably Figure 3b; line 288 “de” should be “the”.

Minor errors and typos corrections are needed.

Author Response

(The authors gave the same response as above.)

Reviewer 3 Report

First of all, what is obvious is the fact that the work has a broad description of the components and the communication lines between them. To avoid an article written in extenso, it would be good if the authors focused on a part of the analysis done, in order to leave enough space for the presentation of some concrete aspects of the research.

In paper is using terms of "dark current". Why is so important this current? Why you don't consider white current?  Please give more explanation about some terms used.

Figure 11 is the only about software used, but is too few. It represent only an import library and some methods to configure.  Many captures and explanation of software are more suitable.  It is not necessary to reveal all aspects of programming  but some of them you should.

The histogram in figure 13 is based on figure 12 (USAF pattern)? If yes, please explain the relevance of it. 

It would be good for this article to be restructured to have a cursive and the details provided should be relevant.

Author Response

(The authors gave the same response as above.)

Round 2

Reviewer 1 Report

The revised version addresses all the problems from the review.  There are still minor remarks to design like the need to use an additional 10MHz clock that can be easily derived from ACLK (AXI clock signal). Another problem is using the XADC universal interface over dedicated implementation for obtaining parameters of the chip at work. But those problems are of polemic nature between designers.

The paper brings an interesting implementation case study that is worth publishing.

The paper is written correctly and use apropriate language to presented problems. I can suggest a proof reading of the final version.

Author Response

Dear Reviewer,

Thank you very much again for your comments, we hope that the answers will clarify your doubts.

Reviewer 2 Report

The authors have revised the paper according to the Reviewer's comments. The paper can be accepted after minor revisions. I recommend the authors to proofread the paper for errors and typos. For example: at line 302 it is "permits" and not "permit";at line 325 it is "parameter" and not "parametro"; at line 368 it is not clear what "bake-out to degas" means; at lines 379 "." must be replaced by ","; at line 414 there is a reference to Figure 10 that is probably wrong; at line 491 it is "represents" and not "represent"; at line 528 it is "image" and not "imagen".

Minor revisions are needed to correct some errors and typos as discussed in the comments for authors.

Author Response

(The authors gave the same response as above.)

Reviewer 3 Report

It is an article with a large descriptive volume, but with few results. The general presentation of the idea, the exhaustive presentation of the registers, modules and block diagrams used is of interest, but the part of the results is disproportionately small. In the future, I would recommend that this observation be taken into account. Emphasis must be placed on the results and their interpretation.

Author Response

Thank you very much for your comments. Yes, due to the complexity of the design, it has been a very extensive job and we have probably focused on the more general design details. We will take this disproportionality into account for future work.
